Effect of yeast and essential oil-enriched diets on critical determinants of health and immune function in Africanized Apis mellifera

Canché-Collí César 1
Estrella-Maldonado Humberto 1
Medina-Medina Luis A. 2
Moo-Valle Humberto 2
Calvo-Irabien Luz Maria 1
Chan-Vivas Elisa 1
Rodríguez Rosalina 1
Canto Azucena azucanto@cicy.mx 1
1 Unidad de Recursos Naturales, Centro de Investigacion Cientifica de Yucatan, A.C. , Merida , Yucatan , Mexico
2 Departamento de Apicultura, Campus de Ciencias Biologicas y Agropecuarias, Universidad Autonoma de Yucatan , Merida , Yucatan , Mexico
Harpur Brock
Electronic publication date: 2021 Oct 15
Publication date: 2021
Volume: 9
Electronic Location ID: e12164
Received 2020 Nov 30; Accepted 2021 Aug 25
Copyright: ©2021 Canché-Collí et al.
Copyright year: 2021
Copyright holder: Canché-Collí et al.
License: This is an open access article distributed under the terms of the Creative Commons Attribution License, which permits unrestricted use, distribution, reproduction and adaptation in any medium and for any purpose provided that it is properly attributed. For attribution, the original author(s), title, publication source (PeerJ) and either DOI or URL of the article must be cited.
License URL: https://creativecommons.org/licenses/by/4.0/

Keywords: Honeybee, Nutrition, Immune system, Bee-associated yeasts, Essential oil, Pollen substitutes, Native plants, Native bees

Funding: The CONACYT 219922 The CICY (Fiscal funds) CONACYT scholarship 374928/242994 This work was supported by CONACYT (grant number 219922), and the CICY (Fiscal funds). César Canché-Collí received a CONACYT scholarship (374928/242994). The funders had no role in study design, data collection and analysis, decision to publish, or preparation of the manuscript.

==============================
Nutrition is vital for health and immune function in honey bees (Apis mellifera). The effect of diets enriched with bee-associated yeasts and essential oils of Mexican oregano (Lippia graveolens) was tested on survival, food intake, accumulated fat body tissue, and gene expression of vitellogenin (Vg), prophenoloxidase (proPO) and glucose oxidase (GOx) in newly emerged worker bees. The enriched diets were provided to bees under the premise that supplementation with yeasts or essential oils can enhance health variables and the expression of genes related to immune function in worker bees. Based on a standard pollen substitute, used as a control diet, enriched diets were formulated, five with added bee-associated yeasts (Starmerella bombicola, Starmerella etchellsii, Starmerella bombicola 2, Zygosaccharomyces mellis, and the brewers’ yeast Saccharomyces cerevisiae) and three with added essential oils from L. graveolens (carvacrol, thymol, and sesquiterpenes). Groups of bees were fed one of the diets for 9 or 12 days. Survival probability was similar in the yeast and essential oils treatments in relation to the control, but median survival was lower in the carvacrol and sesquiterpenes treatments. Food intake was higher in all the yeast treatments than in the control. Fat body percentage in individual bees was slightly lower in all treatments than in the control, with significant decreases in the thymol and carvacrol treatments. Expression of the genes Vg, proPO, and GOx was minimally affected by the yeast treatments but was adversely affected by the carvacrol and thymol treatments.

Introduction

One way of improving the health of honey bee colonies is to provide diets containing compounds that strengthen their health and immune function. Beekeepers supply their honey bee colonies with pollen diets or pollen substitutes to strengthen them during critical feeding or illness periods. Research is needed to identify diet components that effectively strengthen honey bee health. Nutritional stress can increase insect susceptibility to diseases, pests, and pesticides (Di Pasquale et al., 2013; Goulson et al., 2015). Nutrition enhances critical determinants of health and immune function in several ways. For instance, dietary protein provides essential amino acids for synthesizing vitellogenin, phenoloxidase and glucose oxidase, and promotes the accumulation of fat body tissue (Alaux et al., 2010; Negri et al., 2019).

The gene Vg is involved in the production of vitellogenin. It is synthesized in insects’ abdominal fat body and has longevity-enhancing, antioxidant and immunological functions in honey bees (Amdam et al., 2004). The proPO gene is involved in the production of phenoloxidase, a precursor (zymogen) of phenoloxidase, which triggers the melanization process, a humoral response to pathogens and parasites (González-Santoyo & Córdoba-Aguilar, 2012). Glucose oxidase is critical to the inhibition of pathogens in bee food. It is produced in the hypopharyngeal glands of worker bees and secreted, through a channel that opens into the mouth onto the hypopharynx, into bee bread and honey (Ohashi, Natori & Kubo, 1999). The amount of fat body tissue is a component in biosynthetic and metabolic activity in bees because it controls the synthesis and utilization of energy reserves and synthesizes most of the hemolymph proteins involved in insect defense against pathogens (Wilson-Rich, Dres & Starks, 2008).

Yeasts isolated from hive products (e.g., bee bread, larval food, and honey; Gilliam, 1979; Rosa et al., 2003) can modulate bee health and immune function. For example, the brewers’ yeast Saccharomyces cerevisiae is widely used in honey bee feed as a source of proteins in periods of nectar and pollen scarcity (Standifer et al., 1960). Although some research has been done on yeasts in bee nutrition, further studies are needed to understand how yeasts isolated from bee food sources modulate colony nutrition.

Essential oils (EOs) are also used as bee diet supplements. Honey bees actively collect several phytochemicals that are also dominant components in essential oils, and many of these compounds have antibiotic activity against pathogens and parasites (Erler & Moritz, 2016). Honey bees fed plant-derived phytochemicals survive longer and have a high capacity to overcome infections (Bernklau et al., 2019). Essential oils or single compounds isolated from them such as camphor, carvacrol, eucalyptol, menthol, thymol, and several sesquiterpenes have been used as topical medication to control Varroa destructor mites and as nutraceutical compounds against Nosema spp. spores (Imdorf et al., 1999; Borges, Guzman-Novoa & Goodwin, 2020). The anti-nosemosis properties of essential oils may result from the lipophilic nature and low molecular weight of their terpens/terpenoids. They can cause cell death or inhibit sporulation and germination of fungi by disrupting the cell membrane structure or inhibiting chitin polymerization of the cell wall (Nazaaro et al., 2017). The objective of the present study was to evaluate the effect of dietary supplements of native bee yeasts and essential oils from a native plant on honey bee health and immunity variables, as a possible strategy for preventing disease in bee colonies. We measured six critical variables in worker bees: survival, food intake, cumulative fat body percentage, and expression of three health- and immune-related genes (Vg, proPO, and GOx).

Newly emerged A. mellifera workers were fed nine artificial diets, five supplemented with yeasts isolated from a native bee Scaptotrigona pectoralis, three with essential oils from Mexican oregano Lippia graveolens, and the control diet; all potentially strengthen health and immune function in A. mellifera. Honey bee health and immune function include complex metabolic pathways; therefore, it is difficult to make assumptions about immunity based on relatively few genes and health determinants. However, the response variables used in this study are critical modulators of health in A. mellifera. Thus, it is feasible to expect that diets containing yeasts or essential oils may enhance worker bee survival, induce higher food intake, cause significant accumulation in the fat body, and produce higher expression of immune-related genes (Vg, proPO, and GOx).

Materials & Methods

Bee-associated yeasts

We extensively sampled yeasts from S. pectoralis bee colonies in proximity to A. mellifera colonies. We isolated and purified all the cultivable yeast strains from the native bee colonies and stored them in 15% glycerol (w/w) at −80 °C, and identified a group of yeasts associated with larval food, bee bread, and honey that are promising supplements in bee nutrition (Lizama, 2011). Yeasts were identified by molecular methods using primers NL1-NL4 and sequencing the D1/D2 domain of the LSU 26S rDNA (Canto, Herrera & Rodríguez, 2017). Of the resulting yeast strain consensus sequences, five strains were selected for use in this study: (1) CICY-RN-358, identified as Starmerella bombicola, isolated from pots containing honey inside nests; (2) CICY-RN-386, identified as Starmerella etchellsii, isolated from pots containing larval food; (3) CICY-RN-413, identified as Starmerella bombicola 2, isolated from pots containing bee bread; (4) CICY-RN-443, identified as Zygosaccharomyces mellis, isolated from pots containing honey; and (5) CICY-RN-Sac2, S. cerevisiae a commercial strain used in diet formulations for feeding bees (see GenBank accession numbers and nucleotide sequences in the Appendix).

Essential oils

Three types of EOs were extracted from L. graveolens. Commonly known as Mexican oregano, this plant belongs to the family Verbenaceae and is distributed throughout the Yucatan Peninsula (Martínez-Natarén et al., 2014). It produces three oil chemotypes called carvacrol, thymol, and sesquiterpenes (Acosta-Arriola, 2011). The EOs were extracted from leaves collected from cultivated crops and wild plant populations following Calvo-Irabién et al. (2014). Chemical characterization of the EOs was done using a gas chromatographer coupled to a mass spectrometer (GC-MS; Agilent Technologies Model 6890N). Chromatographer was equipped with a 5875 B mass selective detector, a splitless injector, and an MS-Chemstation GI701-DA data system incorporating the National Institute of Standards and Technologies (NIST) spectrum library. An apolar column (5% phenyl methylpolysiloxane; Agilent VF-5ms) separated the oil compounds, which were classified into three chemotypes based on the dominant compound: carvacrol, consisting of 66% carvacrol; thymol, containing 49% thymol; and, sesquiterpenes, consisting of a mixture of four sesquiterpenes, β-caryophyllene (14.6 mg, 56%), α-humulene (6.5 mg, 25%), eucalyptol (1.8 mg, 7%) and p-cymene (1.5 mg, 5.9%) (Table 1).

Table 1 Chemical composition of the essential oil of Mexican oregano Lippia graveolens.

Chemical composition and concentration (mg/ml) of three types of essential oils (EO) identified in the Mexican oregano (L. graveolens). The table shows the main compounds of each type of oil produced by this species highlighting (bold) the dominant compounds for each type: carvacrol, thymol, and sesquiterpenes.

Compounds	Essential oil types	
	Carvacrol	Thymol	Sesquiterpenes	
p-Cymene	10.20	6.25	1.53	
Eucalyptol	0.66	1.37	1.88	
γ-Terpinene	2.17	0.74	0.37	
Thymol	2.39	22.07	0.08	
Carvacrol	38.89	2 × 10−7	0.13	
β-Caryophyllene	2.79	6.26	14.69	
α-Humulene	1.32	3.16	6.55	
Caryophyllene oxide	0.47	0.59	0.69	

Diets

Nine artificial diets were formulated for the experiment. The base diet was a simple pollen substitute (base diet) prepared following standard methods for raising adult A. mellifera workers in cages under laboratory conditions (Williams et al., 2013). The base diet was made on a weight/weight basis following the formulation of Van der Steen (2007): soy flour (14%); beer yeast flour (10%); calcium caseinate flour (15%); whey protein flour (4%); sucrose solution (48%, 1:1 w/v in sterile tap water); and linseed oil (9%). We replaced the beer yeast flour with one of five yeast strains isolated from S. pectoralis colonies in the experimental yeast diets. Saccharomyces cerevisiae was included in the feeding experiment because it is widely used in beekeeping for its significant protein and vitamin contributions. Yeasts were freeze-dried before addition to diets and added 7% (w/w) proportion to base formulation resulted in cell density of approximately 539,155 cells/g. This cell density is similar to the observed in the sources from which yeasts were isolated. The EOs diets consisted of the base formulation enriched with one of the three L. graveolens chemotypes, classified according to their dominant phytochemical compound, and contained no added yeast material. The EOs were added to the base formulation at a 1% (w/w) proportion.

Bees and experimental design

The bee source were 20 colonies of Africanized A. mellifera bees reared from five-frame nucleus colonies in the experimental apiary of the Faculty of Veterinary Medicine and Zootechny (Facultad de Medicina Veterinaria y Zootecnia) of the Autonomous University of Yucatan, Mexico (20°51′51.62″N; 89°36′45.35″W). This apiary is surrounded by 28 hectares of dry tropical forest containing at least 90 nectar- and pollen-producing plant species. The regional climate is subtropical with a 25 °C average annual temperature, 948 mm annual precipitation and an elevation of 8 m.a.s.l. During summer 2018, 1–2 frames containing worker pupae near emergence were extracted from each of twelve colonies with sufficient worker brood frames (3–4 frames). These extracted frames were incubated in constant darkness at 36 °C air temperature, 70% relative humidity, and constant airflow to remove excess CO2 until adult workers emerged. Newly emerged workers were collected after 48 h and placed in groups of 24–30 bees inside individual cages (plastic containers, 11.0 cm × 7.5 cm × 9.0 cm). Cages were randomly assigned to one of the following treatments: (A) S. bombicola diet; (B) S. etchellsii diet; (C) S. bombicola 2 diet; (D) Z. mellis diet; (E) S. cerevisiae diet; (F) carvacrol diet; (G) thymol diet; (H) sesquiterpenes diet; and the base diet only (control). Cages were kept in constant darkness at 36 °C air temperature, 70% relative humidity, and constant airflow to remove excess CO2 throughout the experiment. Diets were supplied at 0.85 ± 0.4 g per cage every three days, and feeders containing old food replaced every three days to avoid fermentation or microbial degradation. In addition, fresh, sterile water (∼2.0 g) was supplied daily to bees.

For each treatment, three cages were assigned to measure survival, another three measure fat body accumulation, another three measure gene expression, and food intake was measured in all cases as described below (Fig. 1). The diet was the main factor in the experimental design, and we made paired comparisons between the control diet and the target diet (yeast or EO) to test the effect of the diet. Time (day) and hoarding cages were set as random variables to estimate the variance due to intrinsic differences within the factors of days and cages.

Survival

Cages assigned to measure survival were monitored for 12 days. Estimates of bee survival per cage in the corresponding treatment were calculated by the number of living and dead bees per cage per day during the 12-day experiment. These data were used to construct survival probability curves and calculate median survival time, that is, the shortest time for which bee survival was less than or equal to 0.5. Dead individuals were removed daily to prevent contamination.

Food intake

For all cages in all treatments, the average daily food intake per bee (% food intake/bee/day) was measured over nine days (Williams et al., 2013). Feeders were filled with the respective diet type, weighed (initial mass), and placed in the corresponding cage. The time (h) each feeder was placed in a cage and the number of alive bees in the cage (initial bees) were recorded. After three days, the feeders were removed, recording the time (final time) and the number of living bees in the cage (final bees). Feeder weight (final mass) was measured and the total amount of food consumed was calculated by subtracting final mass from initial mass. Average food evaporation was estimated by placing control feeders in cages under the same conditions as the treatment feeders but without bees. Food intake was calculated by subtracting the average evaporation value from the observed differences in mass in the experimental diets. The time each feeder was in a cage was calculated by subtracting the initial time from the final time. Hourly intake by cage was calculated by dividing food intake mass by feeder time in the cage, and hourly intake per bee was calculated by dividing hourly cage intake by the final number of living bees in a cage. Daily bee intake was calculated by multiplying hourly bee intake by 24.

Fat body

Accumulated fat body percentage per bees was measured following Wilson-Rich, Dres & Starks (2008). During nine days, at three-day intervals, a group of three bees was removed from each cage. Bees were cold anesthetized, killed by separating the head from the abdomen, and desiccated for three days at 24 °C in silica gel. The abdomens were weighed individually (initial mass) on an analytical balance (Ohaus™; 0.001 g accuracy), and soaked in diethyl ether for 24 h to dissolve all fatty tissue. They were dried for three days and weighed (final mass). Fat body proportion per bee was calculated by dividing final abdomen mass by initial abdomen mass.

Vg, proPO, GOx gene expression

Expression of Vg, proPO, and GOx in caged bees was measured by RT-qPCR. During nine days, three bees were removed every three days from each cage, killed and dissected. Expressions of Vg and proPO were measured using abdomens and GOx expression using heads. The abdomens and heads were each pooled by cage, ground and homogenized in TRIzol® reagent (Invitrogen, USA) following manufacturer instructions for total RNA isolation. Genomic DNA contamination was extracted from the samples using DNAse I (DNA-free kit, Ambion, USA), and RNA concentration and purity measured with a NanoDrop® ND-1000 spectrophotometer (Thermo Scientific NanoDrop Technologies, LLC, Wilmington, DE, USA). Quality of the RNA was checked by electrophoresis in agarose gel (1.5%). A well-established, in-house protocol was used to produce single-strand complementary DNA (cDNA) from 300 ηg purified RNA of each sample by reverse transcription using a final concentration of 50 U/μL MultiScribe™ Reverse Transcriptase (Invitrogen/Life Technologies, CA, USA). Synthesis of cDNA was done at 42 °C for 50 min. The first-strand cDNA was analyzed by RT-qPCR using primers for Vg, proPO, GOx, and the RPS5 reference gene. A 150-bp Vg amplicon was generated with NM_001011578.1-F (5′-GTTGGAGAGCAACATGCAGA-3′) and NM_001011578.1-R (3′-TCGATCCATTCCTTGATGGT-5′) (Tsuruda, Amdam & Page, 2008). We developed an enzyme-specific primer pair for the present study and used it to generate a 130-bp proPO amplicon with the Primer Express ver. 3 software (Applied Biosystems, Foster City, CA, USA): AY242387.2-F (5′-GAACGGCTATGTAATCGTCTTGGA-3′) and AY242387.2-R (3′-TACCGCTGGGTCGAAATGG-5′). A 201-bp GOx amplicon was generated with AB022907.1-F (5′-GAGCGAGGTTTCGAATTGGA-3′) and AB022907.1-R (3′-GTCGTTCCCCCGAGATTCTT-5′) (Yang & Cox-Foster, 2005). One pair of internal reference primers was used for generating a 115-bp amplicon of ribosomal protein S5 (reference gene RPS5): XM_006570237.2-F (5′-AATTATTTGGTCGCTGGAATTG-3′) and XM_006570237.2-R (3′-TAACGTCCAGCAGAATGTGGTA-5′) (Evans, 2006). Analysis by RT-qPCR was run with a thermocycler StepOne™ Real-Time PCR System using the StepOne ver. 2.3 software (Applied Biosystems). Each sample per cage was run in three internal replicates, which were analyzed with RT-qPCR using a 48-well PCR plate. Amplification of cDNA was done using SYBR Green Master Mix as a detection signal (Applied Biosystems), 2 μL cDNA, and 4 μM of each gene-specific primer at a 0.4 μM concentration in a 20 μL reaction volume. Amplification runs were: 95 °C for 3 min; and 40 cycles of 95 °C for 30 s, 58 °C for 30 s and 72 °C for 1 min. The specificity and accuracy of the RT-qPCR products for all samples were verified with melting curve analysis. Relative expression levels (REL) of the Vg, proPO, and GOx genes were calculated using the comparative CT (ΔΔCT) method (Schmittgen & Livak, 2008). The relative quantification of the target genes was normalized to RPS5 expression. The RT-qPCR conditions, following MIQE guidelines, are described in the Appendix.

Figure 1 Experimental design.

Two brood frames were removed from 20 colonies of Africanized worker bees Apis mellifera and incubated under controlled laboratory conditions. The newly emerged bees were placed in hoarding cages and fed one of nine artificial diets for 9–12 days. Next, six response variables indicative of the bees’ health and immune function were measured in fed bees (illustration and photograph credit: Azucena Canto and Luis A. Medina-Medina).

Data analysis

Data from Day 0 are shown in the figures but were excluded from statistical analyses. On Day 0, the first dose of food was provided at different times (1–3 h), but feeding was done within a 30-min margin of difference during the remainder of the experimental period. The data were tested for normality (Kolmogorov–Smirnov test), homogeneity of variance (Levene test), and transformed if needed before statistical analysis. The worker bee survival curve and median survival time were compared between the control (base diet) and each diet treatment using the non-parametric Kaplan–Meier survival analysis and the Log-Rank test. Survival curves for each treatment were built with the survival (Therneau, 2019) and the survminer (Kassambara & Kosinski, 2019) packages in the R software (R Core Team, 2018). In each treatment, the Kaplan–Meier estimator was applied to calculate the probability that an individual bee would survive at a particular time. The Log-Rank test compared survival probability in a specific treatment with survival time in the control.

To statistically test the effect of the diet on food intake (square-root transformed), fat body accumulation (not transformed), and expression (REL Log10-transformed) of the Vg, proPO, and GOx genes, we used Linear Mixed Effects Models (LMM) and planned contrasts to test specific hypotheses about differences between the control and each of the diets (Schad et al., 2020). Analyses were conducted in the lmerTest package (Kuznetsova, Brockhoff & Christensen, 2019) in the R System for Statistical Computing (R Core Team, 2018). The maximum likelihood fitted the LMM with the lmer function, and the sum of squares of Type III for unbalanced designs was used to calculate F and p values. The Satterthwaite’s method was applied for approximating effective degrees of freedom. Diet type was established as a fixed factor and day and cage as random variables. The analysis was based on the following model with no interaction: response variable = diet type + (1—day) + (1—cage). Orthogonal contrasts were applied to compare each of the eight diets with the control using the contest function. Dunnett test was used to compare each treatment to the control, and the Kenward-Roger’s method to approximate degrees of freedom.

Results

Survival

Although worker bee survival in all treatments had lower survival probability curves (Fig. 2) than in the control, the curves did not statistically differ (Kaplan–Meier, Log-Rank test, p > 0.05). However, median survival time (black dashed line, Fig. 2) did tend to be lower in bees fed the carvacrol (median = 3 days) and sesquiterpenes (median = 3 days) treatments than in those fed the control (median = 4 days).

Figure 2 Survival probability.

Survival curves for Africanized worker beesApis mellifera fed one of nine artificial diets for 12 days. Enriched diets contained one of five bee-associated yeasts or one of three types of essential oil: (A) Starmerella bombicola; (B) Starmerella etchellsii; (C) Starmerella bombicola 2; (D) Zygosaccharomyces mellis; (E) Saccharomyces cerevisiae; (F) carvacrol; (G) thymol; (H) sesquiterpenes. Each plot compares the control (base) to each treatment diet. Dashed black lines indicate median survival time. P-values are derived from the Log-Rank test to compare differences between the control diet and each treatment group. Data points depict the mean value of survival probability.

Food intake

Diet type affected average daily food intake per bee (LMM, F8,288 = 2.54, p = 0.0109, Fig. 3). The estimation of random variables identified a greater variation between days but lower variation between cages (Day = 37.01%, Cage = 15.58%). Food intake was higher in the Z. mellis (mean ± SD = 1.34 ± 1.48%, post-hoc Dunnett test, p = 0.0089) and S. cerevisiae (mean ± SD = 1.41 ± 2.52%, post-hoc Dunnett test, p = 0.0456) diets than in the control. In the other yeast and EOs treatments food intake was similar to the control (mean ± SD = 0.76 ± 0.48%, post-hoc Dunnett test, p > 0.05).

Figure 3 Food intake.

Food intake for Africanized worker bees Apis mellifera fed one of nine artificial diets for nine days. Enriched diets contained one of five bee-associated yeasts or one of three types of essential oil: (A) Starmerella bombicola; (B) Starmerella etchellsii; (C) Starmerella bombicola 2; (D) Zygosaccharomyces mellis; (E) Saccharomyces cerevisiae; (F) carvacrol; (G) thymol; (H) sesquiterpenes. Each plot compares the control (Base) to each treatment diet. Data points depict the mean ± SE of food intake (mg/bee/day).

Fat body

Fat body accumulation was significantly affected by diet type (LMM, F8,270 = 3.08, p = 0.0024, Fig. 4) and variation due to days and cages was relatively lower (Day = 1.28%, Cage = 5.80%). In the yeast diets, fat body accumulation did not differ from the control (mean ± SD = 50.10 ± 17.15%, post-hoc Dunnett test, p > 0.05), but two EOs treatments had a significant negative effect on fat body accumulation compared to the control: thymol (mean ± SD = 32.68 ± 12.60%, post-hoc Dunnett test, p = 0.0001) and sesquiterpenes (mean ± SD = 38.11 ± 14.90%, post-hoc Dunnett test, p = 0.0373). Fat body accumulation in the other yeast diets and in the carvacrol diet did not differ from the control (mean ± SD = 43.02 ± 13.83%, post-hoc Dunnett test, p = 0.3280).

Figure 4 Fat body accumulation.

Percent of fat body accumulation for Africanized worker bees Apis mellifera fed nine artificial diets for nine days. Enriched diets contained one of five bee-associated yeasts or one of three types of essential oil: (A) Starmerella bombicola; (B) Starmerella etchellsii; (C) Starmerella bombicola 2; (D) Zygosaccharomyces mellis; (E) Saccharomyces cerevisiae; (F) carvacrol; (G) thymol; (H) sesquiterpenes. Each plot compares the control (Base) to each treatment diet. Data points depict the mean ± SE of fat body accumulation in bee abdomens (%).

Expression levels of the Vg, proPO, and GOx genes

Gene expression was highest on day six for Vg, proPO, and GOx, both in the yeast and EOs treatments. Diet affected expression of the Vg gene (LMM, F8,78 = 18.04, p < 0.0001, Fig. 5) and variation was substantial between days but null in response to cage (Day = 96.15%). Compared to the control, Vg expression was lower in the carvacrol (mean ± SD = 17.96 ± 12.68 REL, post-hoc Dunnett test, p < 0.0001) and thymol treatments (mean ± SD = 17.03 ± 9.72 REL, post-hoc Dunnett test, p < 0.0001). In contrast, the relative expression of the Vg gene was similar to the control in the yeast diets and the sesquiterpenes diet (mean ± SD = 27.76 ± 17.29 REL, post-hoc Dunnett test, p > 0.05).

Figure 5 Expression of Vg gene by RT-qPCR.

Relative expression levels (REL) of the Vg gene in Africanized worker bees Apis mellifera fed nine artificial diets for nine days. Enriched diets contained one of five bee-associated yeasts or one of three types of essential oil: (A) Starmerella bombicola; (B) Starmerella etchellsii; (C) Starmerella bombicola 2; (D) Zygosaccharomyces mellis; (E) Saccharomyces cerevisiae; (F) carvacrol; (G) thymol; (H) sesquiterpenes. Each plot compares the control (Base) to each treatment diet. Data points depict the mean ± SE of Vg REL.

Diet affected expression of the proPO gene (LMM, F8,78 = 14.68, p < 0.0001, Fig. 6) and variation in gene expression was greater between days and negligible between cages (Day = 85%). Expression of the proPO gene in the S. bombicola (mean ± SD = 8.41 ± 6.37 REL, post-hoc Dunnett test, p = 0.0224), carvacrol (mean ± SD = 5.83 ± 3.27 REL, post-hoc Dunnett test, p < 0.0001) and thymol (mean ± SD = 7.20 ± 4.26 REL, post-hoc Dunnett test, p = 0.0013) treatments was lower than in the control. In the other yeast treatments and in the sesquiterpenes treatment gene expression was similar to the control (mean ±SD = 12.44 ± 8.90 REL, post-hoc Dunnett test, p > 0.05).

Figure 6 Expression of proPO gene by RT-qPCR.

Relative expression levels (REL) of proPO gene in Africanized worker bees, Apis mellifera, fed nine artificial diets for nine days. Enriched diets contained one of five bee-associated yeasts or one of three types of essential oil: (A) Starmerella bombicola; (B) Starmerella etchellsii; (C) Starmerella bombicola 2; (D) Zygosaccharomyces mellis; (E) Saccharomyces cerevisiae; (F) carvacrol; (G) thymol; (H) sesquiterpenes. Each plot compares the control (Base) to each treatment diet. Data points depict the mean ± SE of proPO REL.

Diet affected expression of the GOx gene (LMM, F8,78 = 22.73, p < 0.0001, Fig. 7) and variation in gene expression was higher between days and negligible between cages (Day = 83.60%). The yeast diets S. bombicola (mean ± SD = 7.89 ± 3.88 REL, post-hoc Dunnett test, p = 0.0046) and S. etchellsii (mean ± SD = 8.10 ± 4.43 REL, post-hoc Dunnett test, p = 0.0090) resulted in a moderate reduction in gene expression, but the carvacrol (mean ± SD = 4.23 ± 2.06 REL, post-hoc Dunnett test, p < 0.0001) and thymol (mean ± SD = 2.79 ± 1.76 REL, post-hoc Dunnett test, p < 0.0001) treatments resulted in the strongest negative effect on gene expression compared to the control. In the rest of the yeast diets and in the sesquiterpenes diet, GOx expression was similar to the control (mean ± SD = 11.38 ± 6.28 REL, post-hoc Dunnett test, p > 0.05).

Figure 7 Expression of GOx gene by RT-qPCR.

Relative expression levels (REL) of GOx gene of Africanized worker bees, Apis mellifera, fed nine artificial diets for nine days. Enriched diets contained one of five bee-associated yeasts or one of three types of essential oil: (A) Starmerella bombicola; (B) Starmerella etchellsii; (C) Starmerella bombicola 2; (D) Zygosaccharomyces mellis; (E) Saccharomyces cerevisiae; (F) carvacrol; (G) thymol; (H) sesquiterpenes. Each plot compares the control (Base) to each treatment diet. Data points depict the mean ± SE of GOx REL.

Discussion

Survival

The low worker survival observed in all the diets, including the control, suggests an undocumented factor that probably shortened lifespan in the caged bees. One possible cause is the amount of protein in all the diets. Nitrogen-rich diets (protein or amino acids) have deleterious effects on survival, resulting in surprisingly high mortality in caged honey bees (Pirk et al., 2010; Archer et al., 2014; Gregorc et al., 2019). The underpinning mechanism for this is unclear, but nitrogen-rich diets may increase oxidative stress by overproducing reactive oxygen species (ROS). Diets with a 1:5 protein:carbohydrate ratio strongly reduce survival in caged honey bees (Archer et al., 2014), and the ratio in the present study was approximately 1:1; this is a large amount of nitrogen for bees and may explain the low survival rates. Further research directly addressing the harmful effects of dietary protein on bee lifespan are needed.

The specific yeasts used here exhibited a positive effect on bee survival. This suggests that yeasts naturally associated with bee colony food reserves do not diminish bee health when used as food compounds. When added to pollen, brewers’ yeast (S. cerevisiae) positively affects colony-level fitness components such as brood-rearing (Herbert Jr & Shimanuki, 1978; Van der Steen, 2007). Pioneering studies using caged bees found that using brewers’ yeasts in bee food resulted in survival rates similar to those in bees fed pollen (Standifer et al., 1960). Recent studies have found that addition of brewers’ yeasts to pollen substitutes can maintain bee longevity in cage conditions, resulting in survival rates similar to those observed in bees fed bee bread (Smodiš Škerl & Gregorc, 2014; Esanu, Radu-Rusu & Pop, 2018; Amro, Younis & Ghania, 2020). These studies and our results suggest that the nutrients provided by bee-associated yeasts are a good food source for bees and could further improve bee health.

Except for the sesquiterpenes treatment, the EOs treatments negatively impacted worker bee survival. The carvacrol and thymol treatments may have been somewhat toxic to the bees since they are xenobiotics and are not naturally present in bee diets. These results coincide with previous findings indicating that diets containing carvacrol and thymol reduce survival in honey bees (Maistrello et al., 2008; Borges, Guzman-Novoa & Goodwin, 2020). Moreover, high mortality rates have been reported in caged bees in response to increases in carvacrol and thymol concentrations in feeding treatments (Glavan et al., 2020). Carvacrol and thymol are promising monoterpenoids as treatments against Nosema fungi and parasitic Varroa mites in honey bee colonies (Imdorf et al., 1999; Borges, Guzman-Novoa & Goodwin, 2020). Therefore, evaluating their effects on bee survival is essential to quantify their toxicity (Ebert et al., 2007; Gashout & Guzmán-Novoa, 2009; Glavan et al., 2020). For example, thymol is approved in Europe (EC, 2007) for controlling V. destructor in bee colonies because it has low toxicity in evaporation treatments. However, feeding and topical assays are needed to construct a complete toxicity spectrum. There is no current evidence of the harmful effects of sesquiterpenes on bee survival. The thymol and carvacrol concentrations in the sesquiterpenes diet were much lower than in the other two EOs treatments.

Based on the present results, bee-associated yeasts seem to be suitable natural elements for sustained laboratory feeding of bees, with no negative impacts on survival. However, using EOs from L. graveolens in bee diets requires more research to evaluate the effect of different concentrations in feeding experiments under laboratory and colony conditions.

Food intake

Under laboratory conditions, worker bee food intake can decrease in response to handling practices, stress due to confinement and artificial feeding (Even, Devaud & Barron, 2012). However, in the present study, food intake was similar to field conditions, suggesting that the bees ingested sufficient food to maintain their physiological processes.

The high food intake in the Z. mellis and S. cerevisiae diets coincided with vigorous food intake of S. cerevisiae-enriched diets reported in caged bees in another study (Amro, Younis & Ghania, 2020) and was similar to food intake in honey bee colonies under field conditions (Sihag & Gupta, 2011). High food intake of the two yeast-supplemented diets may be due to the nutritional components contained in yeast cells (e.g., lipids or proteins) (Herbert Jr & Shimanuki, 1978) and/or the volatiles produced by microbial metabolism (Christiaens et al., 2014). Food intake in the present EOs treatments was similar to that reported when using different thymol concentrations compared to a base diet of sucrose syrup (Maistrello et al., 2008; Costa, Lodesani & Maistrello, 2010). In another study, food intake containing different concentrations of thymol and carvacrol did not change compared to a control (Borges, Guzman-Novoa & Goodwin, 2020). Therefore, supplementation of bee diets with EOs does not affect palatability.

Fat body

Supplementation with yeasts or EOs did not increase fat body accumulation in bee abdomens, although fatty tissue content remained unchanged throughout the experiment in the yeast treatments. Previous research has focused on evaluating how pollen, the source of protein and lipids for bees, can influence bees’ accumulation of fatty tissue. For instance, diets containing pollen tend to increase fat body reserves more than sugar-rich diets (Barragán, Basualdo & Rodríguez, 2016) and variations in pollen quality can affect the fat body. For example, in one study, fatty tissue was higher in bees fed polyfloral pollen than in fed monofloral pollen, despite similar feed intake (Alaux et al., 2011).

The present results indicate that the yeast supplements helped to maintain fat body mass constant. Yeast-enriched formulations contain the amount of lipids and proteins needed for a constant fat body accumulation (Di Pasquale et al., 2013). Additionally, the higher food intake observed in two of the yeast treatments can provide the bees a surplus of nutrients for their metabolism, allowing a constant fat body accumulation, which is critical since caged honey bees can experience a drastic reduction in abdominal fat tissue (Toth et al., 2005).

Our results confirm previous reports of the pre-existing fat body reserves in newly emerged bees (i.e., Day 0; Ramsey et al., 2019), probably accumulated during the larval and pupal stages. Furthermore, most examined bees exhibited similar fatty tissue percentages, indicating that they had the resources required to produce anti-pathogen molecules immediately after emerging (Wilson-Rich, Dres & Starks, 2008).

Expression levels of Vg, proPO, and GOx genes

The expression of the Vg, proPO, and GOx genes was relatively unaffected in the yeast treatments but reduced in the carvacrol and thymol treatments. The Vg expression pattern did not differ between the diet types, even at the highest levels at day six. This kind of Vg profile is to be expected during the first week of development in nurse bees because they need a higher amount of vitellogenin at this life stage to synthesize royal jelly to feed the brood and queen (Bitondi & Simões, 1996). However, vitellogenin synthesis can be nutritionally regulated using protein-rich diets, thus, the expression of the Vg gene. Although pollen and bee bread are good diets for stimulating Vg expression and titers (Cremonez, De Jong & Bitondi, 1998; Di Pasquale et al., 2016), pollen substitute diets containing native yeasts can also provide this stimulation. For example, bees fed a diet containing yeasts and soybean exhibited Vg levels similar to bees fed bee bread (Cremonez, De Jong & Bitondi, 1998). However, in another study, bees fed yeast and soybean supplemented diets had lower Vg levels than naturally fed bees in a colony (Bitondi & Simões, 1996). In addition, bees fed a fermented pollen substitute containing bee bread-associated microorganisms (bacteria and yeasts) promoted Vg levels similar to those in bees fed on bee bread, indicating that bee-associated microbes can help to enrich artificial food (Dias et al., 2018). Although further research is needed, the present results suggest that bee associated-yeasts may be a safe protein source for maintaining healthy Vg levels and a promising pollen substitute in artificial diets.

We observed that supplementation with carvacrol and thymol induced down-regulation of Vg. Its expression is known to be low in adult bees exposed to thymol vapors (Boncristiani et al., 2012), and expression is delayed in bee larvae fed diets containing thymol (Charpentier et al., 2013). Alterations in Vg expression are critical to bees because this protein influences many life-history traits and is connected to immune function via nutrition (Amdam et al., 2004). Negative impacts on Vg expression may explain the low median survival time observed in the carvacrol treatment since reduced Vg levels result in a reduced lifespan. In addition, changes in hormone metabolism may be responsible for Vg down-regulation in response to thymol exposure (Boncristiani et al., 2012). Further data on carvacrol or sesquiterpenes on Vg expression are needed because these EOs components are valuable for controlling pathogens in apiculture (Borges, Guzman-Novoa & Goodwin, 2020).

Regardless of diet type, the overall pattern of proPO expression differed from that of Vg and GOx, exhibiting a gradual increase until day six with a slight reduction by day nine. In worker bees, proPO expression reaches a plateau within the first week of adult life and remains similar from nurse to forager stages, with very slight variability (Lourenço et al., 2005; Schmid et al., 2008).

Immune function in insects is costly in nutrients and energy, dependent on protein intake (Lee, Simpson & Wilson, 2008). Since phenoloxidase in bees is an enzyme (i.e., protein), this same dynamic can be expected in the proPO (gene-enzyme) system. However, there are no previous reports of an evident influence of nutrition on proPO expression in bees. However, proPO expression is positively influenced by protein-rich diets in model insects such as Spodoptera littoralis and Anabrus simplex (Srygley et al., 2009; Cotter et al., 2019). Of the yeast treatments, only S. cerevisiae caused any up-regulation of proPO expression. This is not entirely unexpected since this yeast has long been marketed as a high protein- and vitamin-content dietary supplement for bees (Standifer et al., 1960). The carvacrol and thymol treatments induced lower proPO expression. Expression of proPO in adult worker bees is linked to dietary diversity (Alaux et al., 2010), but EOs are xenobiotics and not part of the natural bee diet. This suggests that EOs may act on gene expression in other ways, possibly blocking additional activation factors needed to regulate proPO expression tight or activating detoxification pathways (Berenbaum & Johnson, 2015). The proPO system is the first line of humoral response in bees, and it regulates cellular immune responses. In the present study, the reductions in gene expression caused by carvacrol and thymol could increase individual susceptibility to pathogens such as Nosema and viruses, eventually weakening the entire colony. Even though carvacrol and thymol reduce proPO expression, they have also been shown to reduce Nosema spores (Maistrello et al., 2008; Costa, Lodesani & Maistrello, 2010; Borges, Guzman-Novoa & Goodwin, 2020).

GOx expression patterns were initially undetectable in emerging bees in all the treatments, but gradually increased with age. This coincides with previous findings that GOx expression and glucose oxidase secretion in hypopharyngeal glands are age-dependent and attain the highest levels in old nurse bees and foragers (Takenaka et al., 1990). In addition, expression of GOx may be nutritionally dependent (Bucekova et al., 2014); for instance, GOx levels are higher in bees fed polyfloral pollen than those fed monofloral or sugar diets (Alaux et al., 2011). This is supported by research using a Spodoptera exigua model in which high GOx levels were only observed in insects fed an artificial diet (Merkx-Jacques & Bede, 2005).

The yeast treatments slightly negatively affected GOx, although different cell densities or yeast strain cocktails may produce different responses. There is no research to date on bee-associated yeast in pollen substitutes as nutritional modulators of GOx. However, one study did find that the hypopharyngeal glands, the main GOx synthesis site (Ohashi, Natori & Kubo, 1999), were stimulated in bees after being fed a pollen substitute containing the yeast Candida tropicalis (Amro, Younis & Ghania, 2020).

Both the carvacrol and thymol diets caused reduced GOx expression. The observed low GOx expression was probably due to a toxicity effect from carvacrol and thymol, which can be expected considering these compounds act as xenobiotics. Diets with added EOs containing carvacrol or thymol as dominant compounds may trigger detoxification mechanisms in bees. Detoxification would limit the resources available for GOx transcription. Further evaluation is needed to confirm this possibility by testing the responses of variables related to xenobiotic detoxification pathways in honey bees (Berenbaum & Johnson, 2015). Any possible adverse effect of carvacrol and thymol on GOx levels deserves further attention since it can negatively affect bee capacity to sterilize brood food and honey, potentially making entire colonies more susceptible to pathogen attack.

Conclusions

Although the present results do not conclusively demonstrate that diets enriched with bee-associated yeasts directly benefit bee health, they do show they can maintain bee survival, are palatable to bees, and maintain fat body accumulation in bees. Furthermore, diets enriched with yeasts can keep immune-related genes within an acceptable and optimal range of expression; for instance, adding S. cerevisiae to the bee diet can upregulate proPO expression. Future research on yeast-supplemented diets needs to explore different cell densities and yeast cocktails and evaluate yeasts’ antagonistic potential towards intestinal pathogens such as Nosema spp. Even when the EOs diets did not significantly decrease bee survival and food intake, the notable adverse effects of the carvacrol and thymol treatments on fat body accumulation and expression of the Vg, proPO, and GOx genes suggest that these EOs can imprint stress on bees, possibly affecting bee health. Our findings encourage further analysis of EOs under laboratory and in-hive conditions to identify and quantify their impacts, and adjust their application and management in beekeeping to enhance honey bee health. Our study contributes to understanding native yeasts and EOs as nutritional modulators of bee health as strategies for improving health and immune function in A. mellifera colonies.

Supplemental Information

Supplemental Information 1 Nucleotide sequences

Nucleotide sequences of Starmerella bombicola (437bp, 358, B), Starmerella etchellsii (393bp, 386, C), Starmerella bombicola 2 (388bp, 413, D), Zygosaccharomyces mellis (542bp, 443, E), and Saccharomyces cerevisiae (519bp, Sac2 brewing strain, F).

Click here for additional data file.

The authors thank Rodolfo Tut, Víctor Iuit and Filiberto Bacab for technical assistance in the apiaries; Luis Simá and Gabriel Dzib for sampling and collecting plant material to obtain essential oils; Matilde Ortíz for technical support in the molecular biology laboratory; and Dr. Edén Magaña-Gallegos for support in fabricating environmental controllers for the experimental area. We thank the anonymous reviewers, Daniel Borges, and Jorge Navarro for their careful reading of the manuscript and their many insightful corrections, comments, and suggestions, especially in statistics.

Additional Information and Declarations

Competing Interests

Author Contributions

DNA Deposition

Data Availability

The authors declare there are no competing interests.

César Canché-Collí and Humberto Estrella-Maldonado conceived and designed the experiments, performed the experiments, analyzed the data, prepared figures and/or tables, authored or reviewed drafts of the paper, and approved the final draft.

Luis A. Medina-Medina, Humberto Moo-Valle and Elisa Chan-Vivas performed the experiments, authored or reviewed drafts of the paper, and approved the final draft.

Luz Maria Calvo-Irabien performed the experiments, prepared figures and/or tables, and approved the final draft.

Rosalina Rodríguez conceived and designed the experiments, performed the experiments, prepared figures and/or tables, authored or reviewed drafts of the paper, and approved the final draft.

Azucena Canto conceived and designed the experiments, performed the experiments, analyzed the data, prepared figures and/or tables, authored or reviewed drafts of the paper, and approved the final draft.

The following information was supplied regarding the deposition of DNA sequences:

The yeast strains tested are available at GenBank: MW267941 to MW267945.

The following information was supplied regarding data availability:

The data is available at CICY Repositorio-Gobierno de México: Canto A., Canche-Colli, C., Estrella-Maldonado, H., Rodríguez, R. 2020. Evaluation of artificial diets in Apis mellifera 2018-2020. CICY Repositorio-Gobierno de México. https://cicy.repositorioinstitucional.mx/jspui/handle/1003/1753.

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
