# Peer review of "Effect of yeast and essential oil-enriched diets on critical determinants of health and immune function in Africanized Apis mellifera"

_PeerJ, doi:10.7717/peerj.12164_

## Round 0.1 · original submission · Major Revisions

Thank you for your manuscript. It has been reviewed by three experts in the field. Reviewers 1 and 3 felt the manuscript needed additional clarity throughout and especially in the presentation of gene expression results. Reviewer 2 had some major concerns about the survival data you have presented and the relevance of some of your assays.

I agree with the assessments put forward by the reviewers. I ask that you carefully address each of their points and submit a revised copy.

Reviewer 1 ·

Basic reporting

Overall, the article was written in clear, unambiguous English.

1. Minor grammar revisions are needed (e.g. lines 24, 47, 69-70, 71-73, 103, 110-111, 176-178, 206-209, 350).
2. I suggest using ‘honey bees’ instead of ‘honeybees’ throughout the manuscript (Snodgrass, 1956).
3. Line 60. Please consider including the reference for using yeast of S. pectoralis (Lizama, 2011), and perhaps use ‘bees’ instead of ‘honey bees’.
4. In line 72, please list the four ‘informative variables’.
5. Please consider the use of a specific stage of the bee development instead of ‘broods’, such as ‘worker pupae’ (line 154).
6. I assume the authors refer to the role of glucose oxidase in social immunity, but the term ‘community defense function’ (line 197) is unclear, it would be helpful if the authors rephrase or define the concept.
7. The authors use the term ‘house keeping primer’, I suggest following MIQUE guidelines (Bustin et al., 2009) and consider the term ‘reference gene’ (line 223, 230-231).
8. Lines 287-289. Please check the use of brackets; the paragraph is unclear, please revise structure.
9. In line 337, the authors used the term ‘nutritional status’, however, this variable was not previously defined in the manuscript, it would be helpful if the authors define the concept or use the term “food intake”.
10. In line 413, it is unclear if by ‘markers’ you refer to gene expression.
11. As a minor comment, it would be helpful if the authors present the scientific name the first time that the organism appears in the manuscript and use the abbreviation or common name after. For example, the scientific name of Apis mellifera is in line 370 but its firs mention in in line 45, ‘native bees’ are mentioned in line 87, but their scientific name was included until line 95, and the scientific name of the brewer’s yeast is specified in line 380 but previously mentioned in line 344.
12. Also, I suggest revising the nomenclature of genes; italicize throughout the manuscript and be consistent to avoid confusion (e.g. line 497 had the name of the genes -or proteins- and not the italicized abbreviations.)

The authors provided information and references in the background section to show context. However, I would suggest revising the first paragraph of the manuscript and include additional information on why nutrition is important for immune responses and an appropriate reference (lines 46-48).

Also, it would be clearer for the reader if the authors use one definition for your variables. For example, you are using ‘critical markers’ and ‘critical regulators’ as synonyms, and ‘defensive traits’, ‘immune traits’, 'immune defensive traits’ and ‘social defensive characteristics’ interchangeably, which makes the reading hard to follow. ‘Defensive traits’ and ‘critical markers’ are in the title, the authors are not defining or listing the traits or markers in the manuscript clearly.

Experimental design

The authors described the methods in detail, there is enough information to replicate the study. The research question is well defined. However, I recommend revising the following:

1. The reference of the software used to design the primers mentioned in line 226.
2. Please include the primer concentration (line 237) and the reference of the supplementary excel file, which complements the information of the section ‘Gene expression: RNA Extraction, First-Strand cDNA Synthesis, and qPCR’.
3. Schmittgen & Livak (2008) reported a method to calculate relative gene expression by normalizing the target gene expression to a reference gene and a calibrator. It would be helpful if you report the relative fold change of the treatment groups compared to the control in the ‘Results’ section, and not only the relative expression normalized to the reference gene (lines 301-303).
4. Please consider removing the information in lines 198-210, it is the rationale for using those genes in the study but it’s not part of the ‘Materials and Methods’ section, the authors could incorporate the information in the ‘Introduction’.
5. In the supplementary excel file, please include the units for food intake, and define the treatment key (A-I).

Validity of the findings

The statistical analyses seem appropriate, but I suggest the following:

1. To include if the data was tested for normality and homogeneity of variance before subjecting it to a mixed model (1merTest, which to the best of my knowledge is an ANOVA).
2. In the 'Results' section, please include degrees of freedom and the sample size (lines 277-281, 288-290, 296-301, 307-313, 322-327).
3. The authors report the variance withing each treatment (lines 281-285, 290-295, 303-306, 318-320, 329-33); if the authors want to sustain that there are is no significant variability between groups, I recommend considering an appropriate analysis, such as Levene’s test.
4. Honey bee immune responses are complex, and is difficult to make assumptions on individual and social immunity based on the expression of three immune related genes and two heath markers (survival and food intake). I suggest revising lines 88-90 to acknowledge the limitations of the study. The approach of the discussion is in line with the scope of the study.

Additional comments

The authors provide valuable information on the effect of diets enriched with bee associated yeasts (Starmerella bombicola, Starmerella etchellsii, Starmerella bombicola 2, Zygosaccharomyces mellis, and Sacaromyces cerevisiae) and essential oils of Lippia graveolens ( carvacrol, thymol, and sesquiterpenes) on health markers (survival, food intake, fat body percentage) and the expression of immune related genes (Vitellogenin, prophenoloxidase, and glucose oxidase) in honey bees. The article provides enough background and context to understand the research gap. Also, the methodology and statistical methods seem appropriate, and the discussion is in line with the scope of the study. However, I suggest including more information on the effect of nutrition on honey bee immunity in the ‘Introduction’ section; following the MIQUE guidelines to report the methods and results of the gene expression analysis; revising the gene nomenclature throughout the manuscript; using an appropriate statistical analysis to report differences in variability between groups; and to use the same definition for the measured variables throughout the manuscript.

I hope the authors find my suggestions helpful,

Annotated reviews are not available for download in order to protect the identity of reviewers who chose to remain anonymous.

Reviewer 2 ·

Basic reporting

see below

Experimental design

see below

Validity of the findings

see below

Additional comments

The manuscript entitled “Effect of yeast and essential oils-enriched diets on critical markers in the immune system and defensive traits of Africanized Apis mellifera” aimed to analyze the effects of the dietary supplementation with native yeasts or EOs on workers bee in lab conditions. For that, worker bee survival, fatty body mass, and gene expression of vitellogenin (Vg), prophenoloxidase (proPO), and glucose oxidase (GOx) were measured to evaluate the effect of additive components.
Several native yeast strains obtained from Scaptotrigona pectoralis, not from Apis mellifera as is suggested by the authors, and three chemotypes of Mexican oregano, Lippia graveolens that differ in their main dominant component were tested.

In addition to the minor comments attached below, I have two main concerns that arise from the review of this manuscript:

First, I think the assays with yeast are independent of the EOs assays. The aims are completely different and the possible conclusions are unrelated to each other.
When yeasts are added to a normal bee diet as a supplement it is expected that the metabolism and immune system of bees remain optimal as with a normal diet. Maybe an immune challenge can be expected? In this case, the response variables chosen by the authors were correct to test that aim. However, when phytochemicals (as EOs) are added to a normal diet as an additive is expected that this compound acts as a xenobiotic, not as a supplement since EOs are not present in the diet naturally. In that case, to be metabolized after its ingestion, EOs possibly induce detoxification mechanisms. Thus, the variable response to be tested should be related to the search for transcripts associated with detoxification pathways, for example, not immune genes. (Berenbaum & Johnson. 2015. Xenobiotic detoxification pathways in honey bees. Current Opinion in Insect Science).
Moreover, for the determinations tested I think that the sampling size for fat body mass (three bees/replica every three days) and gene expression (three bees/replica pooled sampled every three days) are very low. Due to the high costs of molecular determinations, in a future experiment, authors could choose to include more individuals (without pooled) in fewer sampling times to increase the reliability of the results obtained.

Second, the results obtained on bee survival. All treatments were made maintained worker bees in captivity for 12 days. This is a habitual procedure in honey bee research. It is normal and desirable mortality lower than 20% in the control group. In the present research, mortality obtained in all assays is abnormally high and I think that some of the measured variables are inconsistent (as Fat body scarcely developed in mature bees) due to this.
In control treatments, mortality rates should not be higher than 10-20 %. If this not happens, maybe some extra factor is influencing. Authors should check and rule out some factors before continuing or repeating these tests as the assays temperature range, humidity conditions, ventilation of the device, food availability, among others. For example, 36 C is an optimal temperature to achieve the emergence of bees from a capped comb but, for bioassays in devices with little ventilation, it is advisable not to exceed 32 degrees.

Specific comments:

1. The section must be sorted in descending order:
- Honeybee immune system (emphasizing the components that will be measured in this work)
- Bee nutrition
- Yeast background
- EOs background
- Aim/hypothesis/highlight

2. The aim is not well state

3. In M&M section minimize the background.

4. The groups were statistically compared with the control group, not with each other. Multiple comparisons were not made? Please, inform the N evaluated in each response variable.

5. The figures are many and individual (I suggest to group all yeast and all EOs). In addition, they do not show the statistical differences found nor the N. Authors should present figures with legend to be self-explanatory.

·

Basic reporting

The following are some minor edits that should be done, mostly to help with readability of the manuscript:

-You use a combination of ‘honeybee’ and ‘honey bee’ in your manuscript – for consistency, just use one. I would say ‘honey bee’ as two words is used a bit more commonly in the literature, but either is fine. Just be consistent (you use ‘honey bees’ on lines 57, 405, and in Figure 1; the rest of the time you use ‘honeybee’).
-Somewhat awkward wording on line 75 with “fatty bodies”. Maybe “accumulated fat bodies” or “accumulated fat body tissue”.
-For line 81, you state that a product of the hypopharyngeal glands (glucose oxidase) is secreted via the salivary glands. The reference you cite shows that other enzymes are secreted by both the hypopharyngeal and salivary glands (such as amylase), but that glucose oxidase is secreted solely by the hypopharyngeal gland. Products of the hypopharyngeal gland are not secreted via the salivary ducts, but secreted through a separate channel, that opens into the mouth onto the hypopharynx (see for example, Deseyn and Billen, 2005, in Apidologie: Age-dependent morphology and ultrastructure of the hypopharyngeal gland of Apis mellifera workers (Hymenoptera, Apidae)). That line should be reworded slightly.
-In the title of the manuscript, there should not be an ‘s’ in oils in “essential oils-enriched diets” – it should be “essential oil-enriched diets”.
-Line 144 needs the word ‘it’: “…feeding experiment because it is widely used…”
-Line 153 should be reworded to: “…and an elevation of 8 m amsl.”
-Line 176 should say “filled”, not “willed”.
-Line 230 should “One pair of internal housekeeping primers was used…
-Line 243 should say “Relative expression levels… were calculated…”
-The Data Analysis sections should ideally state the Type I error rate used to determine significance.
-Line 338 sounds awkward, and should be reworded. Maybe the authors meant to keep it as one sentence: “It can be reasonably assumed therefore that yeasts naturally associated with bee colony food reserves will not diminish the nutritional status or survival of bees when used as food compounds, AS the specific yeast cells density used here exhibited an acceptable effect on bee survival.” In addition, “yeast cells density used here” should be “yeast cell densities used here”.
-Line 432 should say “…promissory source in pollen substitute formulas to maintain Vg at healthy levels” not ‘and’.
-The last panel in Figure 1 says 9 replicates / cage / type of diet. According to the Methods, there were 9 diets, but 3 replicates of each cage or diet. So 9 diets x 3 replicates = 27 cages. Not 9 diets x 9 replicates = 81 cages.
-The wording for Figures 3-7 refers to “the experiment in different feeding treatments” instead of listing it out, like in Figure 2. The Figures do go in order, but their titles should stand on their own, in case a reader only looks at one figure in particular. As such, it would be helpful to use the wording “in a nutritional experiment using enriched diets containing one of five yeast species associated with the bee life cycle or three types of essential oil from L. graveolens” in all figures from 2 to 7.

Experimental design

The first point is a minor edit that should be made for clarity; the other points should be addressed, as they are important aspects of the methods that are either not mentioned, or that seem contradictory between what is shown in the Methods, and what is shown in the Results and Figures:

-Survival is calculated as the proportion of living bees over the total number of bees, not over the proportion of dead bees (line 171).
-This could be an issue with either the Methods section, or possibly something that needs to be addressed in the Discussions sections as a finding, but it seems to be an important deviation from what would be expected (unless I am missing something). For the fat body accumulation data, and the expression data for Vg and GOx (though not proPO), why are the starting values at day 0 so different between the control and all the other treatments? There could obviously be a small amount of variation between the bees used for each cage, but the variation shouldn’t be so high, and it shouldn’t be consistently the control that is higher than all treatments, for all three measured parameters. Unless I am misreading the Methods, or it is not mentioned, wouldn’t the day 0 readings all be on bees that had just been put into the cages, that had not begun to feed yet? Were the bees that went into the 8 treatment cages treated / handled differently than the bees that went into the control cages? Again, unless I misunderstood something, all of the bees that went into the cages came from a mix of brood frames from a mix of parent colonies. So why would the immediately-sampled control bees be considerably different than the other treatment bees at day 0? The day 0 data point should be the same across the cages, and should be the base level from which the control and other treatments could then diverge going forward. Or is day 0 supposed to just be the first reading, after 72 h? In which case, it is not day 0, it is day 3, and the experiment was run for 12 days not 9 (though if that is the case, why is there no feed intake value between days 9 and 12?).
-It is not clear how long the experiment was run for. Most of the figures show that the experiment was run for 9 days. However, Figure 2 shows the survival curves and survival data continuing until day 12. Was the experiment terminated at day 9, or day 12? If it was terminated at day 9, why does Figure 2 show three extra days?
-Were cages incubated? Or only frames used to collect newly emerged bees?
-On line 279, it states that feed intake did not differ between EO treatments and the control after 24 h. However, the Methods describe that feed intake was only measured every 72 h, and that is shown in Figure 3. Did the authors mean that there were no differences after 72 h? Or were measurements taken more often than stated in the Methods and shown in Figure 3?

Validity of the findings

Most of the comments here involve explanations and conclusions drawn in the Discussions section that should be re-stated, fleshed out, or backed up better, though a few don’t seem in line with the studies being cited:

-The three studies cited on line 353 are used to suggest different findings than what are found with the current study. However, the three studies support what was found in this study. Borges et al. (2020) found no significant differences in mortality among treatments, though carvacrol, thymol, and oregano oil treatments all had higher mortality than the negative control, similar to the current study. Maistrello et al. (2008) found the same thing in an 11-day study: no significant differences in mortality among treatments, though thymol had higher mortality than the control. Costa et al. (2010) did not show differences in mortality until past day 16, and thymol had higher survival at that point due to its control of Nosema ceranae. I think this section of the Discussion should be re-written to show that the current study in fact supports the findings of these cited studies.
-In the same section as the previous comment, it would be worth citing the work by Gashout and Guzmán-Novoa, 2009, in the J of Apicultural Research: Acute toxicity of essential oils and other natural compounds to the parasitic mite, Varroa destructor, and to larval and adult worker honey bees (Apis mellifera L.), as they looked specifically at what the authors are talking about here on lines 359 – comparing toxicity of essential oils (including thymol and oregano oil, containing carvacrol) on mites, adult bees, and larvae.
-For line 387, it mentions that the study by Borges et al. (2020) showed no significant differences in feed intake between treatments. Again, it should be more explicitly stated that this is also what the current study found – no significant differences in feed intake, but slightly lower with carvacrol, thymol, and oregano oil compared to the control. In addition, I would say that the conclusion that diets containing these compounds are “not terribly palatable to bees” would probably be better stated as “Diets containing thymol and carvacrol appear to be less palatable to bees” based on the findings of this and other studies.
-The conclusions stated in lines 469 to 471 are a bit far-reaching. I think they could be stated alongside evidence that shows the opposite. For example, the studies by Borges et al. (2020), Costa et al. (2010), and Maistrello et al. (2008) already cited in this manuscript show that these essential oils reduce Nosema spores, possibly despite the reduction in proPO expression that they also cause. The authors could comment on how this could be (for example, these EOs may reduce proPO expression, but could have stimulatory effects on other aspects of the honey bee immune response, such as the production of antimicrobial proteins and/or the proliferation and differentiation of haemocytes). That being said, it is still an interesting finding worth noting, as it could increase susceptibility to other pathogens.
-The finding from the study cited on line 488 does not seem very surprising, considering that GOx is produced solely by the hypopharyngeal glands of adult bees. Presumably, its expression would therefore not be affected in honey bee larvae fed different diets, but could be affected in adult bees fed different diets. Instead of suggesting that this perhaps might be the case (lines 488 and 489), the authors should back that statement up by citing the studies already mentioned that showed it was in workers only, and age-dependent (Bucekova et al., 2014; Ohashi et al., 1999; Takenaka et al., 1990).

Additional comments

A very interesting study, and one that is worth publishing. Other than some minor edits for clarity, and better explanation of the findings with regards to other published work, my biggest concern would be the issues with the Methods section, where there seems to be some discrepancies and missing information to explain what the results and figures are showing. In my opinion, those are important issues to be addressed for the paper to be accepted for publication.

That being said, I think the findings are important to the bee research and beekeeping communities, with regards to feed additives that can help to maintain honey bee health parameters in caged settings. The study also points out the need for future research to examine how these feed additives affect honey bee health parameters and survival when fed to bees challenged with pathogens such as Nosema ceranae.

---

## Round 0.2 · Minor Revisions

Thank you for your revision. I have read through the reviewer's responses and yours. I believe we're getting close to an acceptable manuscript. At this point, however, there are many minor points that the reviewers have raised again. As well, there is some concern about your statistical analysis.

Please respond in detail to the comments and return a revised manuscript.

Reviewer 1 ·

Basic reporting

The manuscript has improved, and it is written in unambiguous and professional English. However, I included some suggestions (see comments below).
Line 2. Consider rewording “Nutrition is vital for the health and immune system of honey bees…”
Line 9. Consider rewording “the expression of genes related to the immune system “ or to “the expression of genes related to immune responses…”
Line 23. I don’t think you have to delete the word “genes”. Or you could reword to “Expression of Vg, proPO, and GOx was minimally affected…”
Line 45. I suggest adding a comma and delete “which” to read “…, it is synthesized in the insects’ fact body…”
Line 49. I suggest deleting the word “attack” or reword the sentence.
Lines 51-52. Please revise the use of commas in the paragraph.
Line 72. I suggest using the complete scientific names (Varroa destructor, and Nosema spp). Also, did the essential oils worked against the spores of Nosema? To the best of my understanding, the antifungal properties of essential oils could be related to the lipophilic nature of terpens, and the disruption of the cell membrane during the reproductive state of the fungi, which prevents sporulation (Nazaaro et a., 2017 doi: 10.3390/ph10040086).
Lines 75-78. I suggest using past tense for this section.
Line 98. I suggest deleting “involving multiple variables”. Perhaps the authors mean that the biological pathways associated with immune responses could be assessed using a number of markers. I think that is clear with the explanation in lines 102-106.
Line 105. I suggest substituting “more” with “a”.
Lines 109-112. I suggest revising grammar, the sentence is very long.
Lines 204-205. I assume that the food intake was recorded per cage, and that the food intake per bee (considering live bees) was calculated. Please revise the sentence, it is not clear. Also, the survival was not monitored in all the cages?
Line 208-209. I suggest rewording to “…into the cage and the number of living bees in the cage (initial bees) were recorded”.
Line 2016. Please revise grammar. For example: “The time (hours) that the feeders were in the cages was calculated…”.
Lines 254-257. I suggest: “… and the RNA concentration and purity were measured with a…”.
Line 265. I suggest mentioning the name of the reference gene here.
LIne 287. What do you mean by standardizing? Do you mean that gene expression was normalized to the reference gene, and the control group was used as calibrator? I suggest deleting lines 287-289 to read as follows: “Relative expression levels (REL) of the Vg, proPO, and GOx genes were calculated using the comparative CT (ΔΔCT) method (Schmittgen & Livak, 2008). The relative quantification of the target genes was normalized to RPS5 expression.”
Line 299. I suggest “and transformed if needed…”.
Line 318. Perhaps “effective degrees of freedom”
Line 324. Please revise grammar. I don’t think you needed delete “in the”.
Lines 325 and 328. Please revise this section. If you are comparing the treatments to the control, why are you referring to different days in the parentheses? Also, in the other treatments (yeast) in day 5, the median survival time seems lower compared to carvacol and thymol on day 3.
Line 331. Please revise grammar.
Line 334. Perhaps adding a comma after “three,”
Lines 384-385. Please revise grammar.
Lines 491-494. Please revise grammar.
Lines 501-503 . Please revise grammar.
Lines 534-535. If you found no differences in food intake compared to the control, why do you propose that thymol and carvacol are less palatable?
Lines 53-539. I’m not sure what the authors want to say, please rephrase. Accumulation of lipids in the fat body? How does that connect to your argument?
Lines 540-542. Please revise punctuation marks.
Lines 555-557. What is the difference between the first and second factor? Are the food resources to metabolize different molecules than lipids and protein?
Lines 568-569. Could the increase of Vg expression be related to the age of the bees and not the food source?
Line 572. If you are referring to the protein, then Vg does not need to be italicized.
Line 591. Please revise grammar: “the immune system” or “immune function”.
Line 602-605. If the expression of the genes was compared to the control and if you used the same type of cages, then you can assume that the differences in the gene’s expression were due to the treatments. It would be helpful if you discuss your results considering the control or specify that you are referring to the overall pattern of gene expression.
Line 604-605. Please revise grammar.
Line 625. Please revise this sentence; proPo enzyme also regulates cellular immune responses.
Lines 632-634. This paragraph is redundant, perhaps you can rephrase or delete this part.
Line 625. Please revise the use of punctuation marks.
Line 648. Minor typo: “were stimulated”
Lines 669-670. The names of genes should be italicized.
Lines 671-672. Please consider “possibly affecting bee health and the expression of immune related genes”.
Line 675. Please consider using “laboratory”
Line 677. Is the first time that you mention “chemical stress”. I suggest rephrasing, and perhaps focus on the purpose of enhancing honey bee health.
Lines 679-680. Please consider “antagonistic potential towards pathogens, like Nosema spp”

Experimental design

The authors responded to the first review, the objectives are clear, and the investigation and methods are described in detail. However, I suggest revising how the statistical analysis is being reported (see comment below).

Statistical analysis and results (For example lines 341-344). Perhaps I’m missing something, but I understand that you used a linear mixed effect model to assess the differences between treatments. Normally, you would use the Type III sum of squares to report the F value, DF, and the p value of the model, in this case the effect of the diet, cage, and the interaction cage*day. If the model is significant, then you use the p-values of the post doc analysis for multiple comparisons to interpret your results. Also, you didn’t mention if a correction for multiple comparisons was done for the post hoc analysis. Why are you reporting F values for each of the treatments (diets)? Why are the degrees of freedom different? Where are you reporting the F values, DF, and p values of the model?
Here is an example of a study that used very similar analysis, perhaps you’ll find it useful (see results for oocyte size; Derstine, N. T., Villar, G., Orlova, M., Hefetz, A., Millar, J., & Amsalem, E. (2021). Dufour’s gland analysis reveals caste and physiology specific signals in Bombus impatien s. Scientific reports, 11(1), 1-13. https://doi.org/10.1038/s41598-021-82366-2)

Validity of the findings

The discussion and conclusion are linked to the authors research and findings, and the authors are being critical about the limitations of the study. I included few comments that could improve or help clarify the Discussion section (see comments in previous section).

Additional comments

I have no further comments.

·

Basic reporting

The following are some minor edits that should be done, mostly to help with readability of the manuscript:

-Line 24, the wording around immune system should be changed. Maybe “Nutrition is vital for health and immune function” or “…health and immune system function” or “…health and immunity”
-Likewise, on line 29, “…can enhance genes related to the immune system” or “…genes related to immune function” or “…genes related to immunity”
-Line 39, you need the word genes – “Expression of the genes Vg, proPO, and GOx”
-Line 55, same as lines 24
-Lines 54-56 are a bit too long, and could be broken into two sentences. You could end it at “…determinant of health and immune function.” And start a new sentence “It is synthesized in…”
-Line 57 is awkwardly worded and not very clear. It could be re-worded to something like “The gene proPO is involved in the production of prophenoloxidase, a precursor (zymogen) of phenoloxidase, which triggers the melanization process, a humoral…”
-line 59, remove the word 'attack'.
-line 60, sounds a bit awkward. Could reword to “Glucose oxidase is critical to the inhibition of pathogens in bee food.”
-Line 87, same as line 24
-Line 88, same as line 24
-Line 91, same as line 24
-Line 112, remove the word 'the', just “…known as Mexican oregano”
-Line 113, remove the word 'of', and re-write as “it belongs to the family Verbenaceae”
-Line 158, remove the word 'of', just “to remove excess CO2”
-Line 165, same as line 158, but also remove the word 'the' as well, “to remove excess CO2”
-Line 199, the subheading should include the word 'expression', either “Vg, proPO, GOx gene expression” or “Vg, proPO, GOx expression”
-Line 242, do you mean “…and transformed when needed”?
-Line 265, add 'in the', “…than those in the control (base diet)”
-Line 270, the wording here is awkward, maybe reverse 'daily' and 'consumed', “Average food intake percentage consumed daily by individual caged bees”
-Line 312, you need the word 'treatments' after the brackets, “…in sesquiterpenes (…) treatments than in the control”
-Line 318, same as line 312, you need the word 'treatments' after the brackets
-Line 334, change to “did not differ from the control”
-Line 356, remove the word 'source'
-Line 380, remove the word 'is'
-Line 392-408, in most instances, the word 'feed' should be added before intake “Under laboratory conditions, worker bee feed intake can”. On lines 394, 396, 397, 399, 403, and 405.
-Line 393, add the word 'and', “…stress due to confinement, and artificial feeding”
-Line 436, same as line 24
-Line 459, same as line 24
-Line 461, replace lifetime with lifespan
-Line 493, proPO should be italicized
-Line 498, change “old nurses” to “old nurse bees”
-Line 498, remove the word 'the', “Expression of GOx may be…”
-Line 512, need the word 'considering', “…as is expected considering that these compounds act as…”
-Line 521, reword to: “The adverse effects of the carvacrol and thymol treatments on fat body accumulation, and gene expression of …”
-Line 533, same as line 24
-Figure 4, the title/heading for Figure 4 is a bit awkward. Could maybe be changed to just “Fat body accumulation” or “Percent fat body accumulation”
-Figure 5, in the figure description, Vg should be italicized

Experimental design

no comment

Validity of the findings

The comments here involve a few small things in the Discussion section that should be changed slightly to accurately reflect the studies being cited:

-For line 375, the study by Costa et al. (2010) should be removed, as it did not show lower survival with thymol and/or carvacrol treatment. The other 2 studies (Borges et al., 2020 and Maistrello et al., 2008), however, do show this.
-For the sentence on lines 489-491, your wording suggests that the three sources listed show both a reduction in proPO expression and a reduction in Nosema spores with carvacrol and thymol. However, they only show the latter, while your study showed the former. I would reword this sentence by moving the proPO expression part to the beginning, to make it more accurate: “However, even though carvacrol and thymol reduce proPO expression, Borges, Guzman-Novoa & Goodwin (2020), Costa, Lodesani & Maistrello (2010), and Maistrello et al. (2008) show that they also reduce Nosema spores.”

Additional comments

Other than the minor edits listed here, I think the issues brought up in the first review have all been fixed.

---

## Round 0.3 · Minor Revisions

Thank you for your revision.

I have looked at this revision in light of the previous reviews and do not consider that the reviewers' comments were addressed sufficiently

Could you please thoroughly respond to the requested revisions?

While you made some effort to improve the basic reporting, there is still ample room for improvement, please respond to the reviewer’s requested changes and give the manuscript another check in full.

I would like you to address the concerns about statistics raised by reviewer 2 and the requested changes to the Discussion section.

---

## Round 0.4 · Minor Revisions

Thank you for your revision. It was sent to one expert reviewer to ensure that the requested changes had been made. The reviewer and I are nearly satisfied with the changes and how you have addressed statistical concerns. The reviewer did spot many additional small changes that need to be made and, importantly, some points on how the results are discussed (e.g. between treatment differences). Please make the requested editorial changes and consider the reviewer's requested changes to the conclusions.

Reviewer 1 ·

Basic reporting

The authors responded to the previous suggestions. I included minor revisions for consideration.
Abstract. I think scientific names could be put in parentheses or commas.
In the abstract, the authors mentioned 6 diets with bee-associated yeasts, but there are only 5 listed in parentheses. Perhaps you are considering the control, but you would need to reword.
L57. Correct typo: ‘The proPo gene is involved in the production of prophenoloxidase’.
L66. Are the yeasts isolated from bees, or from hive products (e.g. bee bread, larval food, and honey)?
L84. Perhaps ‘The objective of the present study… ‘
L85. Would it be better to read ‘… of native bee yeasts and essential oils… ‘
L86. Please consider adding a comma after ‘variables’
L136. Please consider adding a comma after ‘diets’
L141. Please consider rewording to ‘The EOs diets consisted of the base formulations enriched with one of the three L. graveolens chemotypes, classified according to their dominant phytochemical compound, and contained no added yeast materials’.
L255. Please revise verb tense ‘We tested..’
L427. I don’t think you can discuss differences between treatments, because you didn’t do multiple pairwise comparisons, please reword. If the authors wish to discuss differences between treatments, I suggest trying a post hoc analysis with multiple comparisons and adjusted p value. It would give more information, type 1 error, and you would be able to compare between treatments, and have a further discussion. You could also include a comparison between yeast treatments and EO treatments. I believe there are some packages that you can use, like lsmeans after running lmer test in R.
L445. I didn’t find Barragan, Basualdo & Rodríguez (2015) in the reference list. I think it was erased from the previous version.
L455-456. I believe previous studies have found that newly emerged bees and adults have fat body tissue, you can check Ramsey et al. (2019; https://doi.org/10.1073/pnas.1818371116). Perhaps a sentence saying that your study confirm previous reports could be included.
L511. Please consider erasing ‘However’.
L518. I don’t think the hyphen is necessary in ’nutritionally dependent’, please verify.
L544. I suggest that the authors revise the ‘Conclusion’ (there is room for improvement); it would be helpful for the reader if the section follows the same order as the manuscript, by pointing out the main and most relevant results on the effect of 1) yeasts and 2) EO on the tested variables. For example, the sentence of lines 544 and 545 is between arguments on EOs.

Experimental design

The authors gave detailed information about the statistical analysis. Overall, the results, discussion, and conclusion are in line with the statistical findings, but please edit L427-428 (see previous comment). Also, to the best of my understanding the Satterthwaite’s method is used to estimate the degrees of freedom, but not the F and P values (it does not adjust the P value). Please revise the sentence in L257-259.

Validity of the findings

The authors gave detailed information on the statistical analyses, they answered the research question, and the discussion is supported by their findings (please see comment about L427-428).

Additional comments

The manuscript greatly improved, and the previous comments were answered. I still have some suggestions, I hope they help.

---

## Round 0.5 · Minor Revisions

Hello, thanks for your re-submission. I read through your responses and I believe there are still some issues arising from the previous review that need to be dealt with.

Statistical analyses have been described such that the authors compare a single control diet to several treatment diets individually using LMM (Line 254). If this is indeed the case, as the previous reviewer described, then the authors need to control for multiple comparisons by correcting their p values in some way (eg. Bonferroni). Alternatively, as the previous review outlined, the authors can run an overall model and follow up with a post-hoc test.

---

## Round 0.6 · Minor Revisions

Thank you very much for your careful re-analysis of the data. I sent this to one reviewer and I read through it myself. The reviewer and I both agree you have handled all the outstanding issues raised.

The reviewer noted a few more grammatical and spelling errors. If you could handle those and give the manuscript one additional, thorough, read I would be happy to accept a resubmission


Again, thank you. I look forward to the next copy.

Reviewer 1 ·

Basic reporting

The authors have responded well to the previous suggestions. I only found minor typos or suggestions:
L260: ‘The analysis was based…’
L263-264: Please revise grammar. For example: ‘Dunnett test was used to compare each treatment to the control, and the Kenward Roger’s method to approximate degrees of freedom.’
L277, 278, 280, 285, 287, 289, 295, 296, 298, 302, 303, 309, 310, 312, 315. Please revise the spelling of ‘Dunnett’ test throughout the manuscript.
L348: ‘to quantify’
L390. I suggest citing the reference before the comma. For example (i.e. Day 0; Ramsey et al. 2019).
L392: ‘… most of the examined bees…’
L401-402. Please revise grammar. For example: ‘Thus, the expression of Vg gene and vitellogenin synthesis can be nutritionally regulated using protein-rich diets’.
L422-424. Please revise grammar and include why further data is needed (to have a structured sentence).

Experimental design

The authors gave detailed information about the statistical analyses and edited the manuscript considering previous suggestions. The results, discussion, and conclusion are in line with the statistical findings.

Validity of the findings

The discussion is supported by the findings of appropriate statistical analyses.

Additional comments

I suggest that the authors do a final check for typos and grammar.

---

## Round 0.7 · accepted · Accept

Thank you for making the requested changes throughout this review process. I see you have satisfied the reviewers and I am happy to accept your manuscript.